# An Interplay between Mitochondrial and ER Targeting of a Bacterial Signal Peptide in Plants

**DOI:** 10.3390/plants12030617

**Published:** 2023-01-31

**Authors:** Tatiana Spatola Rossi, Verena Kriechbaumer

**Affiliations:** 1Endomembrane Structure and Function Research Group, Department of Biological and Medical Sciences, Oxford Brookes University, Oxford OX3 0BP, UK; 2Oxford Brookes Centre for Bioimaging, Oxford Brookes University, Oxford OX3 0BP, UK

**Keywords:** protein targeting, signal peptide, mitochondria, endoplasmic reticulum, tobacco, ER-SURF, plant cells

## Abstract

Protein targeting is essential in eukaryotic cells to maintain cell function and organelle identity. Signal peptides are a major type of targeting sequences containing a tripartite structure, which is conserved across all domains in life. They are frequently included in recombinant protein design in plants to increase yields by directing them to the endoplasmic reticulum (ER) or apoplast. The processing of bacterial signal peptides by plant cells is not well understood but could aid in the design of efficient heterologous expression systems. Here we analysed the signal peptide of the enzyme PmoB from methanotrophic bacteria. In plant cells, the PmoB signal peptide targeted proteins to both mitochondria and the ER. This dual localisation was still observed in a mutated version of the signal peptide sequence with enhanced mitochondrial targeting efficiency. Mitochondrial targeting was shown to be dependent on a hydrophobic region involved in transport to the ER. We, therefore, suggest that the dual localisation could be due to an ER-SURF pathway recently characterised in yeast. This work thus sheds light on the processing of bacterial signal peptides by plant cells and proposes a novel pathway for mitochondrial targeting in plants.

## 1. Introduction

### 1.1. Intracellular Protein Targeting in Plant Systems

Plants are an attractive platform for the production of recombinant proteins due to benefits such as cost-effectiveness, ease of scalability and the lack of potential contaminants present in other systems, such as animal viruses or bacterial toxins [1,2]. Targeting proteins to specific compartments within the plant cell is a strategy often utilised to ensure correct functioning of the recombinant system or increase yields. In particular, the apoplast and chloroplast present the lowest protease activity levels, favouring higher yields due to lower transgene degradation [2]. Retaining or anchoring proteins to the endoplasmic reticulum (ER) has also been reported to increase yields in many cases [3,4]. Plant mitochondria have also been postulated as optimal compartments for the production of heterologous bacterial enzyme complexes in biotechnology [5,6]. Thus, targeting sequences are a crucial aspect of recombinant protein design.

Targeting to the ER or apoplast generally occurs via the presence of an N-terminal signal peptide in the case of soluble proteins or an N-terminal transmembrane signal anchor in membrane-bound proteins. Signal peptides are present in all domains of life and are short sequences with low sequence conservation but a common tripartite structure: the N-terminal region (“n-region”) containing positively charged residues, a central hydrophobic region (“h-region”) and a C-terminal region with polar residues and a signal peptidase cleavage site (“c-region”) [7,8]. In eukaryotes, the signal peptide is recognized by the signal recognition particle (SRP) as it emerges from the ribosome, following which, the ribosome nascent chain complex (RNC) is directed to the ER membrane for translocation through the Sec61 channel [9]. Proteins then enter the secretory pathway, in which they travel from the ER to the Golgi, plasma membrane, extracellular space or the vacuole [10].

Proteins targeted to chloroplasts, mitochondria, peroxisomes and the nucleus generally do not follow the co-translational pathway to the ER and instead are targeted post-translationally from the cytosol to the respective organelles [11]. N-terminal targeting sequences with specific characteristics, such as the transit peptide, pre-sequence, peroxisomal targeting sequence, and nuclear localisation signal, determine the sorting of proteins to the chloroplast, mitochondria, peroxisome and nucleus, respectively [11]. Targeting sequences have low primary sequence conservation and are, therefore, predicted based on their physicochemical properties or tested empirically [12,13]. Certain overlap in targeting signal characteristics can lead to dual targeting to multiple organelles [14]. For example, most dually targeted proteins found in plants are localised to chloroplasts and mitochondria (listed in [14]). This is likely due to similarities in their targeting sequences, in which hydrophobic, hydroxylated and positively charged amino acids are overrepresented, with a low abundance of negative amino acids [13,15]. A lower efficiency of the targeting sequence can also produce dual targeting. For example, studies in yeast found that dually targeted mitochondrial proteins had lower efficiency pre-sequences (defined by combining a set of physicochemical parameters) compared to exclusively mitochondrial proteins [16]. Another study showed that the pre-sequence properties affected the extent of the dual localisation between mitochondria and the cytosol, caused by a differing level of retrograde movement to the cytosol, as shown by exchanging the pre-sequences of aconitase and fumarase in yeast [17].

### 1.2. Role of the ER in Unconventional Targeting Pathways

Some exceptions exist to the predominant targeting pathways. For example, a number of chloroplast proteins are targeted to the plastid via the ER and Golgi bodies [18]. Some peroxisomal membrane proteins are also targeted via an ER-derived compartment [11]. Mitochondrial proteins targeted via the ER have not been reported in plants. However, the early stages of mitochondrial targeting in the cytosol are still not well-known, and several mechanisms are proposed [19]. A novel targeting pathway was recently described in yeast in which mitochondrial precursors are first targeted to the ER surface and then transported to mitochondria [20]. This pathway, termed ER-SURF, involves the yeast Hsp40 Djp1 that is present on the ER surface, to funnel proteins towards mitochondria in cooperation with Tom70 on the mitochondrial outer membrane [19,20]. Some mitochondrial proteins that are targeted towards the ER surface are bound by Get3 [21], some membrane proteins (Oxa1 and Psd1) are recognized by SRP, and other proteins do not have an identified targeting factor for the ER surface [22]. This pathway is also suggested for mammalian cells potentially involving ER-mitochondria contact sites [22]. This raises the possibility as to whether this novel pathway is also present in plants.

### 1.3. The Processing of Bacterial Signal Peptides in Plant Cells

In bacteria, proteins harbouring a signal peptide are targeted to the cell membrane via three different pathways. A major targeting route is the SRP-mediated pathway, which is conserved with eukaryotes. In this route, the emergent signal peptide is recognized by SRP, and the RNC is transported to the SecY channel on the cell membrane (equivalent to the Sec61 channel in the ER of eukaryotes) [23,24]. Alternatively, bacteria use the Sec pathway, which targets proteins to the SecY channel via binding of the signal peptide to SecA, occurring mainly in secreted proteins [23]. Finally, the Tat pathway recognizes signal peptides containing the twin-arginine consensus motif (S/TRRxFLK, where x is any polar amino acid), which transports folded proteins post-translationally through the Tat translocase [23].

Due to conserved features in the structure of signal peptides and in the SRP-mediated pathway, it would be expected for proteins with a bacterial signal peptide to be sorted to the secretory pathway in eukaryotic cells. However, the targeting properties of bacterial signal peptides in eukaryotic hosts, such as plants, are not well-known. An early study reported that a bacterial endoglucanase was secreted by both a prokaryotic and eukaryotic signal peptide in mammalian cells and suggested a similar role for signal peptides in the different domains of life [25]. Contrarily, a study in insect cells showed that a bacterial signal peptide did not efficiently secrete marker proteins, which instead accumulated mostly in the cytosol [26]. In plants, one study reported that the signal peptide from E.coli heat-labile enterotoxin B (LT-B) was capable of directing GFP to the secretory pathway [27]. Expressing bacterial genes in plants has a major scope in biotechnology, for instance, important GM crops, such as those containing herbicide- or insecticide-resistant express bacterial enzymes. More modern crop biofortification and bioremediation projects also express bacterial enzymes in plant systems [5,28,29]. Furthermore, several projects have investigated the production of bacterial and viral antigens in plants for edible vaccines [30,31]. Therefore, further knowledge on the targeting features of bacterial signal peptides in plants would aid the design of recombinant expression strategies.

### 1.4. Targeting of the Bacterial Enzyme PmoB

The bacterial protein PmoB is the largest subunit of the particulate methane monooxygenase (pMMO) complex, an enzyme responsible for the conversion of methane to methanol in methanotroph bacteria [32]. PmoB is an integral membrane protein possessing two transmembrane domains, which separate two large periplasmic domains and contains a cleavable N-terminal signal peptide that directs the protein to the inner membrane in bacteria [33]. The PmoB signal peptide is of interest in regard to the targeted expressing of the pMMO complex in plants for methane detoxification. The PmoB signal peptide is composed of the canonical n-, h- and c-regions, with a signal peptidase cleavage site (AxA), and has not been previously expressed in plant cells, making it an interesting candidate to study the bacterial signal peptide function in plants for biotechnological applications.

Here we show that when expressed transiently in tobacco leaf epidermal cells, PmoB unexpectedly localised to mitochondria. This is most likely due to the native PmoB signal peptide acting as a mitochondrial pre-sequence in plants, differing from its targeting mechanism in bacteria. To further investigate the targeting properties of the PmoB signal peptide, we fused the signal sequence to green fluorescent protein (GFP) and created various signal peptide (sp) mutants. We found that sp-GFP localised to mitochondria and the ER. We identified features characteristic of mitochondrial pre-sequences present in the signal peptide, such as an N-terminal amphipathic helix and a Tom20 consensus motif. However, these sequences alone were not sufficient for mitochondrial targeting. Instead, the presence of a central hydrophobic region, which played a role in ER targeting, was also required for mitochondrial targeting. We thus suggest that targeting in this case could be occurring via the ER-SURF mechanism, in which proteins localise to mitochondria via previous binding to the ER surface. In summary, this work highlights novel aspects of the processing of bacterial signal peptides in plants and the potential of a plant ER-SURF pathway.

## 2. Results

### 2.1. Mitochondrial Targeting of a Bacterial Signal Peptide in Plants

The bacterial enzyme PmoB is targeted to the inner bacterial membrane in methanotroph bacteria. In transient expression in tobacco leaf epidermal cells, PmoB fused to a C-terminal fluorescent protein (PmoB-GFP; Figure 1A) co-localises with rhodamine B hexyl ester [34,35]-stained mitochondria (Figure 1B,C). This differed from the localisation in the secretory pathway, which would be expected if the signal peptide was processed in a similar manner to that in bacteria, as mitochondrial targeting in plants is proposed to occur directly from the cytosol [36,37].

To investigate whether the PmoB signal peptide (sp) alone was responsible for the targeting properties in plants, the sequence was cloned upstream of a GFP moiety (Figure 1D) and transiently expressed in tobacco leaf epidermal cells. This fusion construct (sp-GFP) also localised to mitochondria (Figure 1E,F) in a similar manner to PmoB-GFP indicating that the signal peptide alone is capable of mitochondrial targeting in plants. Interestingly though, most cells expressing sp-GFP also showed a faint labelling of the ER (Figure 1G, Appendix A). ER localisation was confirmed by co-localisation with the luminal ER marker RFP-HDEL [38] (Figure 1G). Controls, including the mitochondrial dye alone and PmoB-GFP with RFP-HDEL, are shown in Appendix A. No clear ER signal can be seen for PmoB-GFP (Appendix A), although this construct has substantially lower expression than sp-GFP (images were taken at 2 days after infiltration as no expression could be seen at 3 days); therefore, ER-localised PmoB could be below the detection limit.

The dual localisation of sp-GFP could be occurring through two different suggested targeting pathways (Figure 1H). In one pathway, the PmoB signal peptide acts as an ambiguous targeting signal capable of dual targeting and directing the protein to either the mitochondria or ER (with a higher affinity for mitochondrial targeting factors, which produces the predominantly mitochondrial localisation). Alternatively, a targeting route that encompasses the dual localisation to mitochondria and ER is the ER-SURF pathway, in which proteins are targeted to mitochondria via previous binding to the ER surface [26].

### 2.2. Sequence Analysis of the Bacterial Signal Peptide Expressed in Plants

As the bacterial signal peptide appears to follow an unusual targeting pathway in plant cells with unexpected mitochondrial localisation, we sought to identify elements in the sequence, which could be responsible for the mitochondrial targeting and dual mitochondrial/ER targeting. Most mitochondrial pre-sequences form an N-terminal amphipathic helix and are rich in positive charges [12,37,39]. Secondary structure prediction using Jpred 4 [40] and PSIPRED 4.0 [41] showed that the PmoB signal peptide features a putative α-helix at its N-terminus and a second longer α-helix further downstream in the sequence (Figure 2A). The N-terminus is rich in basic residues, although two acidic residues are also present, yielding a net charge of +2. The hydrophobic moment (µH) is a measure of the helical amphipathicity. The hydrophobic moment and hydrophobicity were calculated using HeliQuest [42] for each 11-residue window throughout the sequence (corresponding to the size of the first predicted α-helix). The highest hydrophobic moments were found at the very N-terminus of the sequence, and hydrophobicity is higher towards the h-region (Figure 2B). A helical wheel representation shows the opposed distribution of charged/polar residues with respect to apolar residues for the 11-residue N-terminal α-helix (Figure 2C), indicating that this does indeed form a putative amphipathic helix (µH = 0.517).

Many mitochondrial pre-sequences possess a hydrophobic motif with the consensus ϕxβϕϕ (where ϕ is a hydrophobic residue, x is any residue, and β is a basic residue) within a helical structure, which mediates the binding to Tom20 on the outer mitochondria membrane [43,44]. Analysis using MitoFates [45] did not predict the presence of a mitochondrial pre-sequence in sp-GFP; however, it detected two sequences within the signal peptide which conform to the hydrophobic motif structure (LERMA and VGKLL). The LERMA motif is present within the first putative amphipathic helix, and VGKLL is present at the beginning of the second helical domain (Figure 2A). These analyses suggest that the n-region of the signal peptide forms a putative amphipathic helix that may be responsible for the mitochondrial targeting, whereas the more hydrophobic h-region may bind to SRP and be responsible for ER targeting.

To test this, three mutants of the signal peptide were created (Figure 2D): spS-GFP contains a serine in the place of methionine to disrupt the hydrophobic motif, spΔH-GFP is missing the h-region, and spH-GFP has a mutation of serine to isoleucine in the h-region, increasing its hydrophobicity. spS-GFP localised entirely to the cytosol (Figure 2E), indicating that the methionine residue in position 7 and potentially the hydrophobic motif identified, are necessary for both the mitochondrial and ER targeting and not just for mitochondrial targeting as predicted. This suggests that the amino acid composition of the n-region is also a necessary factor for signal peptide recognition by SRP in ER targeting. spΔH-GFP was expected to localise only to mitochondria if the n-region was responsible for mitochondrial targeting and the h-region was responsible for the ER targeting. However, this mutation also localised entirely to the cytosol (Figure 2E). This suggests that the h-region is required for both mitochondrial and ER targeting and that the n-region alone, which contains an amphipathic helix with a Tom20 binding consensus sequence, was not sufficient for the mitochondrial targeting. This is an interesting aspect under the hypothesis of targeting via ER-SURF as the lack of ER targeting capacity, in this case, would abolish the mitochondrial targeting and could not be compensated for by direct targeting to mitochondria. Finally, spH-GFP localised to the ER, indicating that the h-region indeed plays a role in ER targeting. Very faint labelling of mitochondria in spH-GFP could also be observed (Appendix A). Cells expressing spS-GFP, spΔH-GFP and spH-GFP alongside stained mitochondria and RFP-HDEL are shown in Appendix A.

Overall, these data suggest that mitochondrial and ER targeting are interlinked and that both the n- and h-regions contain important information for both ER and mitochondrial targeting. Surprisingly, the h-region is required for mitochondrial targeting in this case, and the marked shift towards ER localisation upon increasing hydrophobicity supports a role of SRP-mediated targeting and the dual localisation with the ER.

### 2.3. Bioinformatic Analysis of Mutants

The deletion of the h-region impaired mitochondrial targeting, which might be due to impairment of a putative ER-SURF pathway or the removal of other mitochondrial targeting motifs, such as the second hydrophobic motif VGKLL, which could be present within the h-region. To test this, we used a bioinformatic prediction analysis of mutants carrying shorter deletions (Figure 3). Predictions were carried out using the subcellular localisation prediction software MULocDeep (https://www.mu-loc.org/; accessed on 1 July 2022) [46]. Control sequences were used to test the prediction capacities of the software. GFP (cytosolic), secGFP (containing a plant secretion signal peptide) and mitoGFP (containing a plant mitochondrial pre-sequence) showed a clear highest probability of localisation to the correct compartment (Figure 3A). PmoB-GFP and sp-GFP showed a highest probability of localisation in mitochondria, also in accordance with empirical data (a close high ER probability in PmoB-GFP is likely due to the presence of transmembrane domains in the full-length protein). Given these outputs, we considered this to be a robust tool to test further mutations in silico.

Mutants containing a series of deletions (spΔ1-4) were analysed in silico (Figure 3B). All deletions showed the highest probability for cytosolic localisation (Figure 3A). spΔ2 and spΔ3 included the second hydrophobic motif (VGKLL), which was absent in spΔH. Interestingly, spΔ4 contained the full n- and h-regions. Therefore, in spite of these regions containing necessary mitochondrial and ER targeting information, the two regions alone are not sufficient for mitochondrial targeting, indicating a role also for the c-region. We next tested deletions of the n-region and found that a full deletion of the n-region (spΔN) yielded a very high probability of secretion, whereas deletion of half of the n-region (spΔN2) yielded a slightly lower probability of secretion with some mitochondrial probability (Figure 3A,B). Finally, a mutant with just the h-region (spjustH) yielded a probability of secretion.

Taken together, this suggests that the h-region and the h- and c-regions combined, are likely able to direct proteins towards the secretory pathway. A combination of all three regions appears to be necessary for mitochondrial targeting. Constructs with shorter deletions in the h-region (spΔ2 and spΔ3) were not predicted to target mitochondria; therefore, it is likely that mitochondrial targeting in the presence of the h-region may be due to ER-SURF as no additional mitochondrial targeting motifs were identified in this region.

### 2.4. Impact of Negative Residues on the Mitochondrial Targeting Capacity of the PmoB Signal Peptide

Mitochondrial pre-sequences have net positive charges, and the presence of negative residues is rare [47]. The PmoB signal peptide contains three negative residues, two in its n-region and one at position -2 in the c-region. We, therefore, asked whether the mutations of these residues would impact upon mitochondrial targeting efficiency and dual targeting. Firstly, glutamate was mutated to arginine at position -2 (spR), as arginine at -2 is very common in mitochondrial pre-sequences [37]. Predictions using MULocDeep showed only a marginal increase in mitochondrial targeting probability for this mutation (Figure 4A,B). On the other hand, the mutation of both glutamate residues in the n-region to alanine was predicted to substantially increase in mitochondrial probability (Figure 4A,B). We thus created the spA-GFP construct and assessed localisation using confocal microscopy. spA-GFP was localised to mitochondria with very little ER signal (Figure 4C).

To test whether the glutamate to alanine mutations in spA-GFP increased mitochondrial targeting efficiency, a quantification of the mitochondrial signal intensity compared to the total cell fluorescence intensity was carried out. The proportion of fluorescence intensity in mitochondria relative to the total fluorescence intensity was calculated for each cell and was significantly higher in spA-GFP (6.5 ± 2.4) compared to sp-GFP (3.5 ± 0.8) (Figure 4D). To confirm that the increase in the proportion of mitochondrial fluorescence was due to a higher contrast with the background and not to more mitochondria being labelled or occupying a larger area, the area of the thresholded mitochondria compared to the total cell area was quantified. No significant differences were found in the proportion of mitochondrial area between spA-GFP and sp-GFP (Figure 4D). This suggested that the mutations of glutamate to alanine improved the mitochondrial targeting efficiency.

## 3. Discussion

### 3.1. Targeting of Bacterial Signal Peptides in Plants

The presence of targeting peptides is a critical aspect in recombinant protein design as it determines subcellular compartments and can affect yield [2]. Here we found that the bacterial protein PmoB, present in methanotroph bacteria, localises to mitochondria when recombinantly expressed in plants. PmoB is the largest subunit of the pMMO enzyme complex, which catalyses the first reaction in methane metabolism in bacteria [48]. Hence, the recombinant expression of pMMO proteins has scope in biotechnology, for example, for methane mitigation [48]. The expression of pMMO in plant mitochondria has been postulated as a potential strategy for the methane mitigation of crop species [6]. Therefore, the expression of PmoB in plant mitochondria may have a biotechnological perspective in the field of methane bioremediation.

The localisation of PmoB in plant mitochondria was unexpected, considering that PmoB possesses a bacterial signal peptide. Even though the SRP pathway, which processes signal peptide-containing proteins, is highly conserved between prokaryotes and eukaryotes, it is not well understood how plants process bacterial signal peptides [27]. Further knowledge in this area could aid recombinant protein design and guide decisions such as replacing a native signal peptide or identifying residues which could increase targeting efficiency. Although replacing a recombinant signal peptide with a plant one is a viable strategy in many cases, sometimes this strategy leads to low yields, as the interaction between the signal peptide and the N-terminus of the protein can affect the translocation efficiency. In some cases, it has been found that using the native signal peptide is the optimal expression strategy [49]. In this work, we showed that the bacterial signal peptide from PmoB does not function like a canonical signal peptide in plants and instead targets proteins to mitochondria. This differs from previous findings that suggest that bacterial signal peptides consistently drive secretion in eukaryotic cells [25,27]. Interestingly, it was recently shown that bacterial TAT signal peptides have the ability to target mitochondria in plants [50]. The PmoB signal peptide does not possess a TAT consensus motif; therefore, it is more likely targeted via the SRP or Sec pathway in bacteria. However, we identified features in its N-terminal, such as a highly amphipathic helix and a Tom20 binding motif, which are seemingly necessary but not sufficient for its mitochondrial targeting properties in plants.

### 3.2. Dual Localisation between the Mitochondria and ER

Dual localisation to mitochondria and the ER was observed when expressing the PmoB signal peptide fused to GFP (sp-GFP). The dual targeting of mitochondrial proteins with another compartment has been described before [51]. In plants, the majority of dual localised proteins are targeted to mitochondria and the chloroplast due to similarities between mitochondrial pre-sequences and chloroplast transit peptides, which are both rich in hydroxylated, hydrophobic and basic amino acids [13,14,51]. Dual localisation between mitochondria and the cytosol is also not uncommon and can be due to a weaker mitochondrial targeting sequence [16] or retrograde translocation into the cytosol driven by folding kinetics or properties of the mitochondrial targeting sequence [17,52]. Dual localisation between mitochondria and the ER has been reported; however, these cases are rare in the literature [14]. It is also unclear how mitochondrial proteins containing N-terminal transmembrane domains or hydrophobic regions in their pre-sequence are not erroneously recognized by SRP [22]. It is thought that a lower hydrophobicity of mitochondrial sequences lowers their affinity for SRP [22]. Moreover, the nascent polypeptide-associated complex appears to co-bind with SRP on the ribosome to avoid non-specific targeting of SRP to the ER [53]. In spite of these mechanisms, some mitochondrial membrane proteins have been found to be SRP clients, and ribosome profiling experiments have found a fraction of mitochondrial proteins to be located on the ER surface [22]. In line with these observations, the ER-SURF pathway has been recently described in yeast, in which proteins are targeted to mitochondria via binding to the ER surface, which could explain the presence of mitochondrial precursors at the ER membrane [20].

### 3.3. The PmoB Signal Peptide might Be Targeting Proteins via the ER-SURF Pathway

The dual targeting observed in this case appears to be interlinked, i.e., full mitochondrial localisation could not be achieved via mutations either empirically or by testing through bioinformatics. It is possible that the signal sequence functions as an ambiguous targeting signal, which binds mitochondrial targeting factors with higher affinity and SRP with low affinity. An increase in hydrophobicity of the h- region in the spH mutant tilted the localisation towards the ER, which is in accordance with an ambiguous targeting signal. However, mitochondrial localisation always seemed to be accompanied by some ER labelling, even when the mitochondrial targeting efficiency was increased by mutating glutamate to alanine residues in the spA mutant. This suggests that the mitochondrial targeting may be dependent on ER targeting, in accordance with ER-SURF as a possible targeting mechanism. The observed labelling of the ER in sp-GFP is very faint, potentially indicating a transient attachment to the ER surface, such as that described in ER-SURF, as opposed to a directed targeting and translocation into the ER.

Finally, the mutants generated also support the idea of an ER-SURF targeting pathway (Figure 5). We propose that the mutation of the hydrophobic motif in the n-region (spS) and the deletion of the h-region (spΔH) could be impeding binding to SRP, thus inhibiting ER targeting and with this the route to mitochondria as no direct mitochondrial targeting is occurring. Moreover, it is possible that the native signal peptide (sp) is recognized by SRP with relatively low affinity, thus allowing a rapid exchange towards chaperones that directly target mitochondria at the ER surface. In contrast, spH has a higher affinity for the SRP, allowing for the translocation of spH-GFP into the ER lumen. The shift in labelling towards the ER with spH supports the idea that SRP was probably binding the native signal peptide in the first place.

Future work studying ER–mitochondria contact sites in plants and putative chaperones that could be involved in an ER-SURF pathway would further elucidate the existence of this targeting pathway in plant cells and its potential role in cell function.

## 4. Materials and Methods

### 4.1. Vector Construction

The PmoB signal peptide from *Methylosinus trichosporium* OB3b (accession number: U31650.2) (DNA sequence: atgaaagctctggaaagaatggccgaactggcgaccggacgggtcggaaagctcctcggcctgagcgttgcggctgcggtcgccgcgacggcggcttcggtggccccggcggaagcg) and its mutants were fused upstream of GFP using PCR (using NEB Q5 high-fidelity DNA polymerase). A vector containing ER-targeted GFP5 was used as template [54]. Bands were extracted from 0.7% agarose gels using NEB Gel Extraction kit and added to a Gateway BP reaction (carried out according to the Gateway^®^ Technology manual [55]). BP reactions were used to transform NEB 5-alpha competent *E. coli* (high efficiency) using heat shock at 42 °C (following manufacturer’s instructions). Entry clones were purified using NEB Monarch Plasmid Miniprep Kit and confirmed using sequencing (Eurofins Genomics). Entry clones were then added to a Gateway LR reaction (following the manual) using the pH7WG2 destination vector for expression in plant cells. Expression clones were used to transform *E. coli* purified using DNA plasmid prep and to transform competent *Agrobacterium tumefaciens* cells (strain GV3101). Approximately 250 ng of DNA was added to 50 μL competent Agrobacterium cell aliquots; cells were incubated on ice for 5 min, subsequently at −80 °C for 3 min and then at 37 °C for 4 min. Next, 1 mL of LB media was added, and cells were incubated at 28 °C for 3 h. Cells were spread on agar plates containing 25 μg mL^−1^ rifampicin and 50 μg mL^−1^ spectinomycin and grown at 28 °C for 3 days.

### 4.2. Transient Transformation of Tobacco

*Nicotiana tabacum* (SR1 cv Petit Havana) plants were grown in greenhouse conditions (21 °C, 14 h light, 10 h dark). Five–six-week-old plants were used for transformation, following the procedure adapted from [56]. In brief, Agrobacterium cultures were spun down at 4000 rpm for 5 min and gently resuspended by pipetting in 1 mL of infiltration buffer (5 mg/mL glucose, 50 mM MES, 2 mM Na3PO4.12H2O and 0.1 mM acetosyringone). A second wash step was performed by repeating the centrifugation and resuspension in 1 ml of fresh buffer. Cultures were infiltrated at OD_600_. A small puncture was made in the abaxial side of the leaf, and the bacterial suspension was introduced using a 1 mL syringe. Infiltrated plants were placed in growth chambers to allow protein synthesis and imaged after 3 days (or 2 days in the case of PmoB-GFP).

### 4.3. Confocal Microscopy

A small leaf piece (approximately 6 mm^2^) was cut with a scalpel and placed on a glass slide with the abaxial side facing up. Images were acquired using a Zeiss LSM800 or a Zeiss LSM880 confocal microscope with an Airyscan detector on a PlanApo 63×/1.46 NA oil immersion objective. Next, 1024 × 1024 images were taken with 2-line averaging. GFP was imaged with excitation at 488 nm and detection at 495–550 nm. Mitochondria were labelled using rhodamine B hexyl ester [34,35]. For this, tobacco leaf pieces were immersed in 1 μM rhodamine B hexyl ester for 15 min, then rinsed in water before imaging. Rhodamine B hexyl ester was imaged with excitation at 561 nm and detection at 570–615 nm.

### 4.4. Subcellular Localisation Prediction

Predictions were carried out using MULocDeep (https://www.mu-loc.org/; accessed on 1 July 2022), a recently developed deep learning-based algorithm, which gives a probability of localisation for 10 different subcellular compartments based on sequence analysis. For mitochondrial prediction, it has been evaluated using over 4000 mitochondrial proteins from three different plant species [46].

### 4.5. Mitochondrial Targeting Efficiency Image Analysis

Total fluorescence was calculated by manually delimiting the cell boundary and measuring the average fluorescence intensity using ImageJ on an 8-bit scale. For mitochondrial intensity, the Yen threshold was applied which segmented mitochondria, and average fluorescence intensity of the segmented areas was measured. Measurements were conducted for a total of 60 cells for each construct, taken from 3 different plants. The proportion of mitochondrial intensity/ total intensity was calculated for each cell. A Mann–Whitney U test was performed to test significant differences between groups. Bar charts with average and standard deviation are shown.

## 5. Conclusions

Overall, this work sheds light on the processing mechanisms of bacterial signal peptides by plant cells. These results show that not all bacterial signal peptides direct proteins to the secretory pathway in eukaryotic cells as was previously postulated, and it opens the possibility of a novel ER-SURF pathway, previously characterised in yeast, to be also occurring in plants. Moreover, some important characteristics of mitochondrial targeting in plants, such as the negative impact of acidic residues at the N-terminus of the pre-sequence, are highlighted. These findings could aid further research involved in recombinant protein design and the compartmentalization of heterologous proteins in a plant system.

## Figures and Tables

**Figure 1 plants-12-00617-f001:**
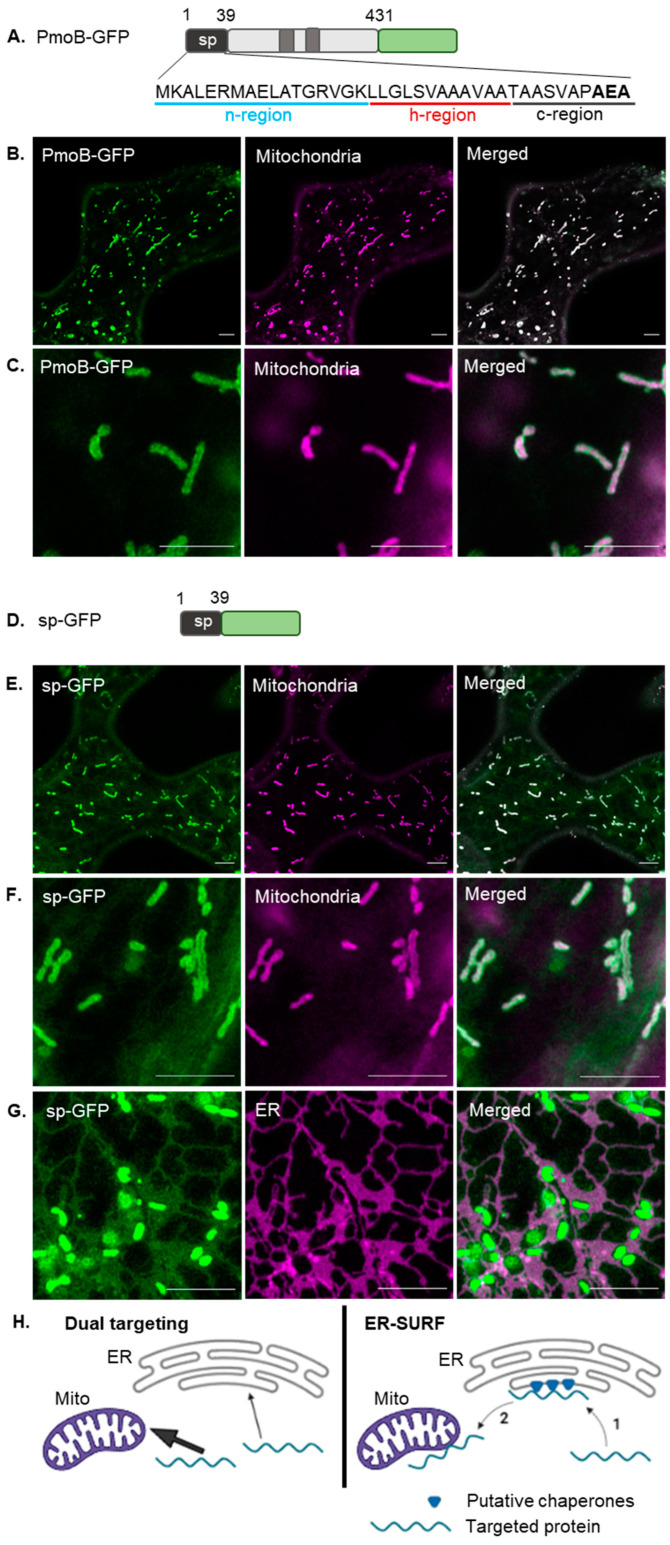
The PmoB signal peptide produces dual localisation to mitochondria and the ER in tobacco cells. (**A**) Diagram of PmoB (in grey) with two transmembrane domains (darker grey squares), comprising a signal peptide between residues 1–39 (black rectangle). n-, h- and c-regions are underlined in blue, red and black, respectively, and the signal peptidase cleavage site is highlighted in bold. GFP (green rectangle) was fused to the PmoB C-terminus. (**B**) Co-localisation of PmoB-GFP with mitochondria labelled with rhodamine B hexyl ester. (**C**) Co-localisation of PmoB-GFP with mitochondria at higher magnification. (**D**) Diagram of sp-GFP, comprising the PmoB signal peptide (black rectangle) fused upstream of GFP. (**E**) Co-localisation of sp-GFP with mitochondria label. (**F**) Co-localisation at higher magnification. (**G**) Co-localisation of sp-GFP with ER marker. (**H**) Schematic diagram for dual targeting and the ER-SURF pathway. For dual targeting, the signal peptide is recognized either by mitochondrial targeting factors or SRP and is targeted to both the mitochondria (Mito) and ER, respectively. A higher affinity for mitochondrial targeting factors confers a predominantly mitochondrial localisation (represented by the thicker arrow). In the ER-SURF pathway, the signal peptide first directs the protein to the ER surface, where chaperones (blue triangles) assist the subsequent funnelling to mitochondria (Mito). Scale bars = 5 μm.

**Figure 2 plants-12-00617-f002:**
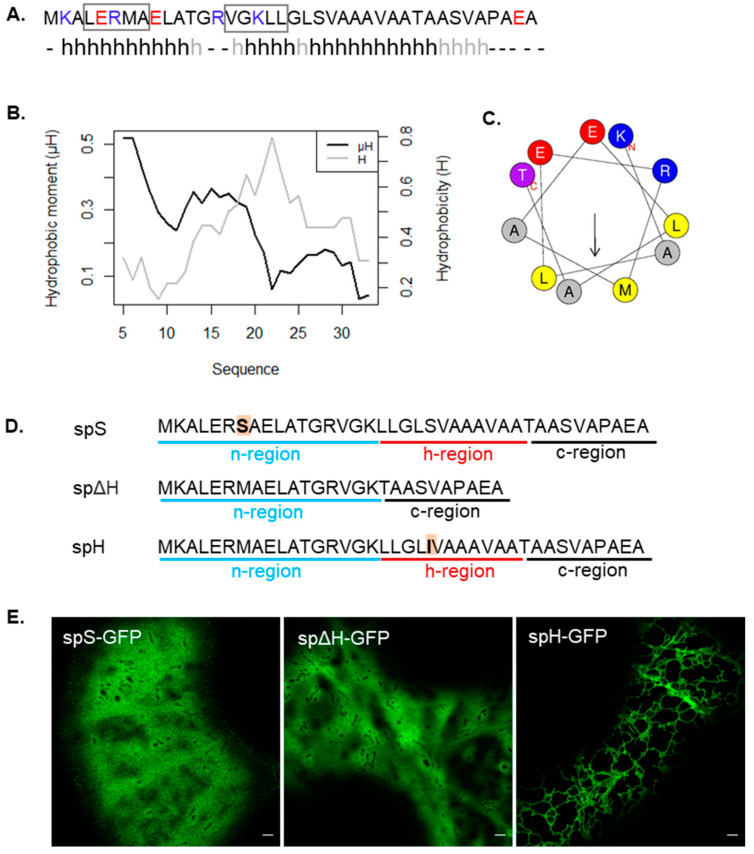
Prediction of an N-terminal amphipathic helix in silico and signal peptide mutations impairing mitochondrial targeting. (**A**) Secondary structure prediction using Jpred 4 and PSIPRED 4.0. An h denotes helical conformation predicted by both software approaches, and light grey h prediction as helix only by PSIPRED 4.0. Basic amino acids are marked in blue, and acidic amino acids are marked in red. The two hydrophobic motifs predicted by MitoFates are highlighted using a rectangle. (**B**) Hydrophobic moment and hydrophobicity of 11 residue windows using HeliQuest. (**C**) Helical wheel diagram of the first 11 residues predicted as an α-helix showing a high hydrophobic moment. (**D**) Sequences of the signal peptide mutants generated. (**E**) Transient expression of mutants in tobacco cells with localisation of spS-GFP and spΔH-GFP to the cytosol, and localisation of spH-GFP to the ER. Scale bar = 2 μm.

**Figure 3 plants-12-00617-f003:**
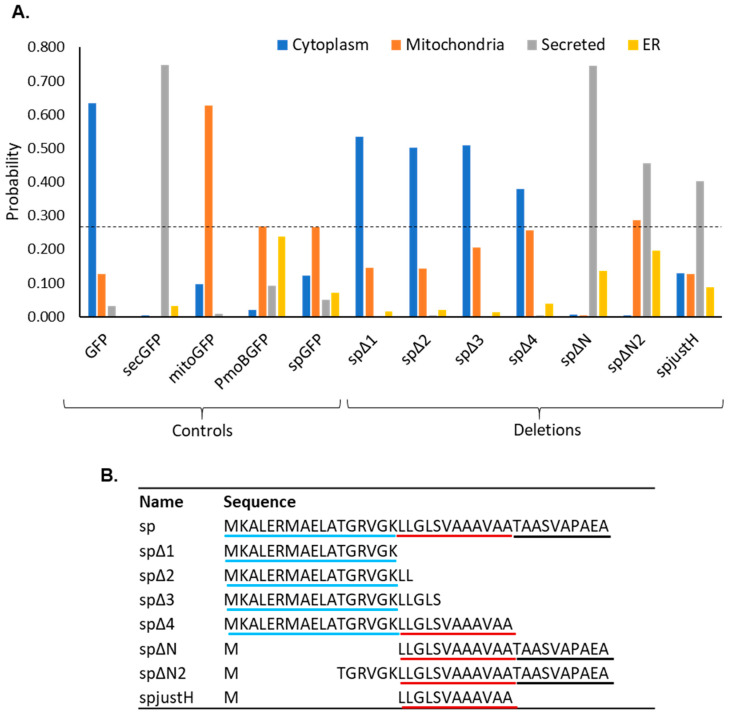
Subcellular localisation prediction of deletions of the PmoB signal peptide. (**A**) Probability of subcellular localisation in each compartment using MULocDeep, for control sequences and deletions of the signal peptide. Probabilities for the four predominant compartments in all cases are shown (cytoplasm, mitochondria, secreted and ER). The dotted line marks the mitochondrial probability of the native signal peptide fused to GFP (spGFP). (**B**) Sequence of the native signal peptide (sp) and all the mutants analysed in MULocDeep. The sequences are aligned to show the position of the deletions; n-, h- and c-regions are underlined in blue, red and black, respectively.

**Figure 4 plants-12-00617-f004:**
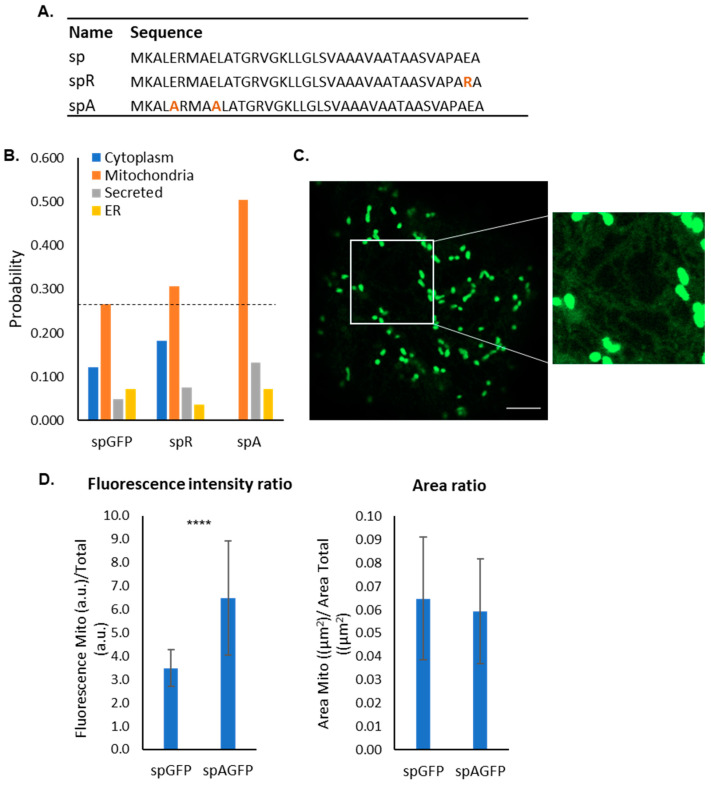
Mitochondrial import efficiency of glutamate to alanine mutations. (**A**) Sequences of spR and spA with mutated residues highlighted in orange. (**B**) Probability of mitochondrial localisation predicted using MULocDeep. (**C**) Example cell expressing spA-GFP showing strong labelling of mitochondria with a low background. Inset shows faint ER labelling. (**D**) Quantification of the ratio of mitochondria to total fluorescence in sp-GFP and spA-GFP (Mann–Whitney U test, *p*-value < 2.2 × 10^−16^, n = 60 cells), and quantification of the ratio of mitochondrial area compared to total cell area of the same group of cells (Mann–Whitney U test, *p*-value = 0.2336). **** = *p*-value < 0.0001. Scale bar = 5 μm.

**Figure 5 plants-12-00617-f005:**
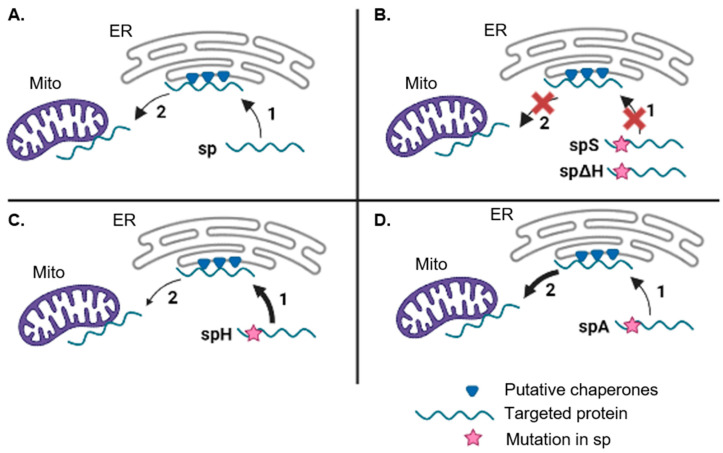
Suggested targeting pathway of the PmoB signal peptide (sp) in plant cells via ER-SURF. (**A**) sp-GFP is first targeted to the ER surface (i.e., by recognition of the signal peptide by SRP) where putative chaperones (blue triangles), such as plant homologues of yeast Djp1, aid in funnelling proteins to mitochondria (Mito). (**B**) Mutations in the hydrophobic motif of the n-region (spS) or deletion of the h-region (spΔH) impair targeting to the ER and thus to the mitochondria. (**C**) Increasing the hydrophobicity of the h-region (spH) produces a larger proportion of precursors at the ER compared to mitochondria, potentially due to higher affinity of the signal peptide for SRP inducing greater translocation of the protein to the ER lumen and vastly reducing its transfer to mitochondria. (**D**) Mutating negative residues in the n-region (spA) increases the efficiency of targeting and translocation into mitochondria following attachment to the ER surface. Targeting efficiency to each compartment is shown according to arrow thickness. Arrow targeting pathway is numbered by order of targeting events.

## Data Availability

Data are included as supporting information. Raw data are available from the corresponding author upon request.

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
