# Peer review of "An Interplay between Mitochondrial and ER Targeting of a Bacterial Signal Peptide in Plants"

_plants, 2023, doi:10.3390/plants12030617_

Round 1

Reviewer 1 Report

The manuscript submitted by Spatola Rossi and Kriechbaumer describes the peculiar fate of a particular bacterial protein subunit upon expression in plant cells and its biotechnological implications. This is likely to occur via a signal peptide, and the features of the unconventional processing of the latter might be a premature evidence for the existence of an ER-surf pathway involved in targeting of proteins to mitochondria in plants. Overall the characterization of the signal peptide is well executed, sufficient theoretical background is provided, and the discussion covers important aspects of most of the findings. I have a few comments and suggestions which are given below:     

An explanation regarding the selection of PMMO subunits for investigation might be useful.

Lines 60-62: The authors might consider expanding a bit and add some characteristic examples.

Line 135, Figure 1: I would add as a control (1) an image showing tobacco leaf cells stained only with rhodamine B 135 hexyl ester and (2) cells co-expressing PmoB-GFP and RFP-HDEL.

Line 149  …with GFP directed primarily to the mitochondria and in a weaker manner to the ER: Please be more specific-this sentence needs to be revised.

Figure 2E: Please provide images showing (1) RFP-HDEL and (2) stained mitochondria in cells expressing the signal peptide variants.  

Lines 243-254: I would shorten the paragraph and transfer the description of MuLocDeep software to the material and methods section. In my opinion it disrupts the flow as is.

Line 297: What about using rhodamine B 135 hexyl ester fluorescence for normalization? I am just wondering whether total fluorescence intensity is a reliable reference for this purpose.

Figure 1H and Figure 5: Labeling of the figures and the respective description in the legends should be improved.

Section 4.1 The section is very poor and needs to be expanded.

Discussion: The authors exclusively discuss the findings regarding the properties of the signal peptide  sequence:

- Any potential significance (and biotechnological perspectives) of PmoB targeting to mitochondria in plant cells?  

- Is there any explanation, why PmoB seems to be present only in mitochondria and not in the ER (Figure1, see also comment for line 135)?

- The authors might consider testing (or explain why they did not use) any of the mutated signal peptide versions abolishing mitochondrial targeting to check the impact on PmoB full length localization.  This would further support the statement made in Lines 117-119.

Supplemental figures and movies mentioned in the manuscript were not available for downloading through the reviewer’s webpage.

Reviewer 2 Report

The manuscript “An interplay between mitochondrial and ER targeting of a bacterial signal peptide in plants” is well structured and the aim of the research work is relevant.  I have the following concerns regarding the current manuscript that authors should consider while revising their manuscript.

1. Statistical analysis is mentioned in the figure only. It should be included in "Materials & Methods".

2. In Figure 4D, What’s the “****”. Please clarify.

3. Provide the reference link (URL) of the software that is used in the manuscript.

4. There are some grammar mistakes in the manuscript. Please check.

Round 2

Reviewer 1 Report

The authors addressed adequately most of my comments and the revised manuscript is considerably improved.